# Endothelial Protein C Receptor and 3K3A-Activated Protein C Protect Mice from Allergic Contact Dermatitis in a Contact Hypersensitivity Model

**DOI:** 10.3390/ijms25021255

**Published:** 2024-01-19

**Authors:** Meilang Xue, Christopher J. Jackson, Haiyan Lin, Ruilong Zhao, Hai Po H. Liang, Hartmut Weiler, John H. Griffin, Lyn March

**Affiliations:** 1Sutton Arthritis Research Laboratory, Kolling Institute, Faculty of Medicine and Health, The University of Sydney, Sydney, NSW 2065, Australia; haiyan.lin@sydney.edu.au (H.L.); rzha9073@uni.sydney.edu.au (R.Z.); helena.liang@sydney.edu.au (H.P.H.L.); 2The Australian Arthritis and Autoimmune Biobank Collaborative (A3BC), Kolling Institute, Faculty of Medicine and Health, The University of Sydney, Sydney, NSW 2065, Australia; lyn.march@sydney.edu.au; 3Versiti Blood Research Institute, Milwaukee, WI 53226, USA; hweiler@versiti.org; 4Department of Physiology, The Medical College of Wisconsin, Milwaukee, WI 53226, USA; 5Department of Molecular Medicine, The Scripps Research Institute, La Jolla, CA 92037, USA; jgriffin@scripps.edu

**Keywords:** skin inflammation, immune cells, mutant activated protein C, protease-activated receptor

## Abstract

Endothelial protein C receptor (EPCR) is a receptor for the natural anti-coagulant activated protein C (aPC). It mediates the anti-inflammatory and barrier-protective functions of aPC through the cleavage of protease-activated receptor (PAR)1/2. Allergic contact dermatitis is a common skin disease characterized by inflammation and defective skin barrier. This study investigated the effect of EPCR and 3K3A-aPC on allergic contact dermatitis using a contact hypersensitivity (CHS) model. CHS was induced using 1-Fluoro-2,4-dinitrobenzene in EPCR-deficient (KO) and matched wild-type mice and mice treated with 3K3A-aPC, a mutant form of aPC with diminished anti-coagulant activity. Changes in clinical and histological features, cytokines, and immune cells were examined. EPCRKO mice displayed more severe CHS, with increased immune cell infiltration in the skin and higher levels of inflammatory cytokines and IgE than wild-type mice. EPCR, aPC, and PAR1/2 were expressed by the skin epidermis, with EPCR presenting almost exclusively in the basal layer. EPCRKO increased the epidermal expression of aPC and PAR1, whereas in CHS, their expression was reduced compared to wild-type mice. 3K3A-aPC reduced CHS severity in wild-type and EPCRKO mice by suppressing immune cell infiltration/activation and inflammatory cytokines. In summary, EPCRKO exacerbated CHS, whereas 3K3A-aPC could reduce the severity of CHS in both EPCRKO and wild-type mice.

## 1. Introduction

Endothelial protein C receptor (EPCR) is a specific receptor for the natural anti-coagulant protein C (PC) and its activated form aPC [1]. It can augment the conversion of PC to aPC, and it mediates most of the anti-inflammatory, anti-apoptotic, endothelial, and epithelial barrier-protective properties of aPC via cleavage of protease-activated receptor (PAR)1 and/or PAR2 [2]. EPCR itself also plays an important role in the regulation of inflammation and stemness. It is the potential stem cell marker for several types of cells [2], including human keratinocytes [3]. EPCR is expressed by most cell types in the human body, such as skin keratinocytes [3] and immune cells [2]. The disruption of *EPCR* gene expression in embryonic stem cells leads to early embryonic lethality in mice [4]. Mice with severe EPCR deficiency are generated by the integration of exogenous DNA elements into the 5′-untranslated region of the *EPCR* gene [5]. These mice develop normally [5] but are more prone to dextran-sulfate-sodium-induced colitis, which causes inflammation and mucosal barrier damage [6]. Conversely, over-expression of EPCR protects transgenic mice from endotoxin-induced injury [7].

Allergic contact dermatitis (ACD) is a chronic inflammatory skin disease caused by repeated exposure to contact allergens, affecting up to 20% of the general population [8]. This condition is characterized by impaired skin barrier and altered immune response [9,10,11]. Although the mechanisms of ACD are well defined [12], the function of EPCR in this disease is unknown.

Recombinant aPC has been shown to have a therapeutic effect in various inflammatory diseases [13], such as allergic asthma [14,15] and mouse contact hypersensitivity (CHS) [16], a commonly used animal model for human ACD [17,18]. However, the anti-coagulant activity of aPC poses an increased risk of serious bleeding. 3K3A-aPC is a mutant form of aPC, which retains the cytoprotective effects of aPC but with less than 10% of its anti-coagulant effects [19,20]. 3K3A-aPC has successfully passed a phase-two trial in patients with ischemic stroke [21], but its function in ACD has not been investigated.

The sub-acute and sub-chronic 2,4-dinitrofluorobenzene (DNFB)-induced mouse CHS model has been shown to better reflect the morphological and functional specifics of chronic ACD. Moreover, this model allows for therapeutic treatment of a pre-established stable and non-necrotic skin inflammation [18]. By utilizing EPCR-deficient (KO) and matched wild-type (WT) mice in this mouse CHS model, the current study aimed to investigate the potential protective effects of EPCR and 3K3A-aPC in this disease.

## 2. Results

### 2.1. EPCRKO Mice Are More Susceptible to DNFB-Induced CHS, and 3K3A-aPC Mitigates CHS

Both WT and EPCRKO mice developed severe CHS symptoms, including ear swelling, erythema, and skin lesions, after nine days of DNFB application (Figure 1A). EPCRKO mice exhibited more severe symptoms than WT, with an approximate 33% increase (*p* = 0.002) in ear thickness at day 14 following DNFB application (Figure 1A). Treatment with 3K3A-aPC significantly reduced ear swelling in WT (*p* = 0.001) and EPCRKO mice (*p* = 0.032), as well as histological features of CHS, such as edema, epidermal necrosis and hyperplasia, and inflammatory infiltration (Figure 1B,C).

Mast cells are one of the main effector cells in ACD [22]. Total and degranulated mast cells were higher in the ear skin of EPCRKO mice compared to WT mice (Figure 1D). 3K3A-aPC inhibited mast cell infiltration and degranulation in WT mice and, to a lesser extent, EPCRKO mice (Figure 1E).

### 2.2. The Expression of EPCR, PC/aPC, PAR1, and PAR2 in Mouse Skin

EPCR, PAR1, and PAR2 are essential receptors for the body’s homeostasis and for aPC to exert its anti-inflammatory and barrier-protective functions. Normal mouse skin expressed PC/aPC, EPCR, PAR1, and PAR2 (Figure 2A). EPCR was primarily located in the basal layer of the epidermis, with selective cells expressing much higher EPCR levels (Figure 2A). This is consistent with EPCR expression in human skin epidermis and suggests that these higher EPCR-expressing cells may be epidermal stem cells, as reported previously [3]. In the EPCRKO skin, PC/aPC expression was significantly increased in the epidermis, particularly in the outer layers; PAR1 and PAR2, especially PAR1, were also markedly upregulated in the epidermis compared to WT skin (Figure 2A,D). However, weak EPCR was present in the epidermis but not the dermis of EPCRKO skin (Figure 2A,D). aPC activity in both the ear skin and flank skin of EPCRKO mice was significantly higher compared to WT (Figure 2E). These data suggest that, intriguingly, the upregulation of PC/aPC and PAR1 may compensate for the deficiency of EPCR in the skin, leading to normal skin function in the unchallenged condition in EPCRKO mice.

The expression of EPCR, PAR1, and PAR2 €n the skin was altered €n the context of CHS. In WT epidermis, EPCR was present in all epidermal layers, with higher levels located in the upper layers rather than the basal layer (Figure 2B,D). The expression of PAR1 and PAR2 was increased, with PAR2 being spread throughout the epidermis but largely located in keratinocytes with nuclei. In EPCRKO epidermis, PAR1 expression was decreased, with most EPCR and PAR2 expression being located in the middle layers of the epidermis—a location where keratinocytes can proliferate and differentiate. The expression of PC/aPC was dramatically reduced (Figure 2B,D). The reduced aPC in EPCRKO CHS ear tissue was confirmed by aPC activity assay, although no difference was found in plasma (Figure 2E). 3K3A-aPC increased the expression of EPCR, PC/aPC, and PAR1, reduced PAR2, and promoted more PAR2 to the uppermost layer of the epidermis in WT mice with CHS (Figure 2C,D). These data provide a clue that lower activity/expression of aPC and PAR1 in EPCRKO skin in CHS may be at least partly responsible for the more severe CHS in these mice. These results imply the upregulation of PC/aPC and EPCR and redistribution/inhibition of PAR2 as a possible mechanism for 3K3A-aPC’s protective role in CHS.

### 2.3. Effect of EPCR Deficiency on Th Cells, DCs, and B Cells

The effect of EPCR deficiency on Th cells, DCs, and B cells was detected by flow cytometry using specific antibody panels. Figure 3D shows the gating strategies for identification of Th1/Th2/Th17/Treg cells. In CHS, EPCRKO skin tissues had higher levels of Th1, Th17, and Treg cells at day 14, while EPCRKO lymph nodes (LN) had lower levels of Th2 cells compared to WT tissues (Figure 3A). No significant difference was found in Th1, Th2, Th17, or Treg cells in blood or spleen between WT and EPCRKO mice (Figure 3A). The effect of 3K3A-aPC on Th cells in skin tissue was examined subsequently. 3K3A-aPC decreased Th1, Th17, and Treg cells while increasing Th2 cells in WT skin, whereas it only decreased Th17 and Treg cells in EPCRKO skin (Figure 3B). To confirm these findings, spleen cells from normal mice were isolated and treated with 3K3A-aPC in vitro for 24 h. In untreated cells, EPCRKO resulted in higher levels of Th2 and Th17 cells compared to WT cells (Figure 3C). 3K3A-aPC decreased Th1 and Th17 and increased Th2 and Treg cells in WT cells, but it only reduced Th17 cells in EPCRKO cells. These data indicate that 3K3A-aPC may require EPCR to exert its function on Th1, Th2, and/or Treg cells.

DCs are well-defined antigen-presenting cells, which display unique characteristics in ACD [23]. Figure 4D demonstrates the gating techniques for the identification of DCs and their sub-sets by flow cytometry. At day 14 of CHS, EPCRKO mice showed an increased number of DCs (CD11c positive cells) in their blood and skin (Figure 4A), along with decreased conventional €DCs and elevated plasmacytoid (p)DCs in LN but decreased pDCs in the skin compared to WT mice (Figure 4A). CD40, CD80, and CD86 are specific for hapten-exposed or activated/mature DCs [24]. EPCRKO DCs expressed higher levels of CD80 and CD86 in the skin and lower levels of CD80 in LN compared to WT cells (Figure 4B). In addition, 3K3A-aPC increased DCs in WT skin by 40% (Figure 4C).

Other immune cells, such as B cells, polymorphonuclear leukocytes (PMN), monocytic cells, and CD3/8 T cells in CHS, were also examined by flow cytometry. The frequencies of B cells, CD3 T cells, natural killer (NK) cells, and PMN were higher, while the frequencies of monocytic cells and CD8 T cells were lower in EPCRKO skin compared to WT (Figure 4D). 3K3A-aPC decreased B cells in the blood and increased them in the skin in WT mice, but it had no effect on these cells in EPCRKO mice (Figure 4E).

These data indicate that EPCR expression affects not only Th cells but also other immune cells in normal and CHS conditions. 3K3A-aPC inhibits Th1/Th17 and stimulates Th2 cells, B cells, or DCs in WT while having little effect on these cells in EPCRKO conditions. This suggests that aPC requires EPCR for its effect on Th1/Th2 cells and DCs.

### 2.4. Inflammatory Cytokines and IgE Levels in Mice with CHS

In CHS, mice with EPCR deficiency exhibited higher levels of IL-6, IFN-γ, TGF-β1, and TNF-α and lower levels of IL-4 in their skin compared to WT (Figure 5). However, there was no significant difference in the plasma levels of IL-4, IL-17, and IL-22 between the two groups. 3K3A-aPC treatment decreased both skin and plasma levels of IL-4 and IL-22, and skin levels of IL-6, IFN-γ, TGF-β1, and TNF-α in WT mice, whereas it only reduced skin levels of IL-22, IFN-γ, TGF-β1, and plasma levels of IL-4 in EPCRKO mice. It is worth noting that there was no difference in either skin or plasma IgE between the two groups, but 3K3A-aPC reduced tissue levels of IgE in WT and EPCRKO mice. These results suggest that increased local inflammatory cytokines/IgE may contribute to more severe CHS in EPCR KO mice. Treatment with 3K3A-aPC mitigates CHS partly via inhibition of these cytokines and IgE.

## 3. Discussion

The study demonstrated that EPCR deficiency exacerbated CHS, and 3K3A-aPC had a therapeutic effect on it in both WT and EPCRKO mice. The presence of EPCR may only partly contribute to the therapeutic function of 3K3A-aPC.

Inflammation and skin barrier destruction are the main features of ACD [25]. The inflammation is regulated by immune cells and their secreted mediators, particularly by T cells and their sub-sets [26,27]. These cells modulate inflammation via the secretion of cytokines such as TNF, INFγ, IL-4, IL-17, IL-22, and TGFβ1. These cytokines were elevated in DNFB-induced CHS skin lesions in all mice, with more severe CHS in EPCRKO mice being accompanied by higher levels of these cytokines (Figure 5). These results indicate that these cytokines are related to the severity of CHS in mice, similar to their effects in patients with ACD [28,29].

High levels of Th1, Th17, and Treg cells in EPCRKO mice with CHS (Figure 3) suggest that alteration in EPCR expression can affect T-cell activation and differentiation. This is consistent with previous studies showing the regulatory role of this receptor in Th 17 cells [30,31]. Furthermore, our results showed that the change in EPCR expression also affected other immune cells, such as mast cells, DCs, and B cells (Figure 1 and Figure 4). Mast cells are both effectors and regulatory cells in ACD [22], and DCs initiate T-cell responses to topically applied hapten in the sensitization phase of CHS [32,33,34]. EPCR deficiency increased the number of DCs in CHS skin and promoted their maturation by increasing CD80 and CD86 expression. Similarly, EPCR deficiency also increased/activated skin B cells. Via their cytokines and autoantibodies [22,35], these immune cells likely promote CHS in EPCRKO mice.

EPCR is a marker for human epidermal stem cells [3]. While no equivalent studies have been conducted in mice, the exclusive expression of EPCR in the basal layer of the mouse epidermis suggests that mouse epidermal stem cells also express EPCR. Deficiency in this receptor can trigger dysfunction of the epidermis, particularly in response to inflammatory stimuli, leading to more severe CHS. Interestingly, EPCR deficiency seems to have organ- and disease-specific effects on inflammation. Deficiency in this receptor protects against bacterial-induced lung injury [36], joint-bleeding-induced inflammation [37], and the development of lupus and anti-phospholipid syndrome [38] in mice. The underlying mechanisms for these organ-/disease-specific reactions are not clear.

When PC binds to EPCR, it can activate aPC [39], providing anti-inflammatory and protective effects to the endothelial and epithelial barrier [16,40]. In this study, no difference in plasma aPC activity was observed between WT and EPCRKO mice under unchallenged and CHS conditions. However, the skin aPC activity and epidermal aPC/PC expression were higher in EPCRKO mice compared to WT mice without challenge (Figure 2). High levels of PC/aPC may compensate for the loss of EPCR, which is essential for skin epidermal function [3,40].

In response to DNFB, EPCRKO mice generated less endogenous aPC within the skin and displayed increased inflammation, resulting in exacerbated CHS. Similarly, EPCR-deficient mice are more susceptible to dextran sulfate sodium (DSS)-induced colitis [6] and experimental autoimmune encephalomyelitis [31]. These results are consistent with those of EPCR point mutation mice, whose EPCR cannot bind PC/aPC [41]. In these mice, thrombotic or LPS challenge results in lower aPC generation, leading to a pro-coagulant/pro-inflammatory status [41]. Therefore, differences in aPC levels/activity between WT and EPCRKO mice under normal vs. CHS conditions may partly explain why we found that EPCRKO mice showed a similar phenotype to WT mice when unchallenged, but in response to inflammatory challenge, they displayed more severe diseases.

The increased presence of PAR1 in the EPCRKO skin epidermis may also contribute to the normal skin structure/function [42], whereas decreased epidermal PAR1 expression in EPCRKO mice in CHS condition may further drive barrier dysfunction and inflammation [43].

3K3A-aPC has normal cell signaling activities but lacks over 90% anti-coagulant activity compared to aPC [19]. Pre-clinical and clinical studies have shown that 3K3A-aPC is at least as effective as aPC in reducing brain injuries caused by ischemic stroke, traumatic brain injury, or amyotrophic lateral sclerosis [13,21,44]. In this study, 3K3A-aPC exerted a therapeutic effect similar to that of aPC in DNFB-induced CHS [16] but at a lower dose, although there has been no direct comparison between the two proteins in CHS. 3K3A-aPC’s diverse pharmacologic benefits are mainly mediated via EPCR, but it also displayed an inhibitory effect on CHS in EPCRKO mice, indicating that this receptor is not fully responsible for 3K3A-aPC’s therapeutical effect on CHS. 3K3A-aPC’s effect on CHS is also likely achieved via stimulation of endogenous PC/aPC and suppression/redistribution of PAR2, as both PAR2 deficiency and aPC attenuate mouse CHS [16]. In contrast, PAR2 is required for aPC’s ameliorating effect on mouse graft vs. host disease [45], indicating the disease-specific functions of aPC-PAR2 signaling pathways.

In summary, this study provides direct evidence that EPCR deficiency can exacerbate CHS and that the administration of 3K3A-aPC can help reduce its severity. These results suggest that EPCR plays a crucial role in protecting against skin inflammation and that 3K3A-aPC could be a valuable therapeutic option for ACD and other inflammatory skin disorders as well.

## 4. Materials and Methods

### 4.1. Animals and CHS Induction

Female C57BL/6J Meox2Cre-EPCRloxP (EPCRKO), aged 6–8 weeks, and their WT littermates were obtained from Kearns Facility, the University of Sydney. EPCRKO mice have normal viability, fertility, and rates of thrombus formation [5]. DNFB (Merck KgaA, Darmstadt, Germany)-induced CHS in these mice (n = 8 mice/group) was generated as described previously [18]. Non-CHS negative control mice (n = 4 mice/group) received the same volume of the solvent (acetone and olive oil: 4:1, *v*/*v*) at the same time points. CHS can be induced in both male and female mice; the female mice were used owing to their less aggressive behavior.

### 4.2. Treatment

3K3A-aPC (1 mg/kg, provided by ZZ biotech, https://zzbiotech.com, accessed on 30 October 2019) or PBS at a volume of 200 µL/mouse were therapeutically administered into mice via intraperitoneal injection daily 1 h before DNFB administration from days 8 to 12. The dose for 3K3A-aPC we selected was based on aPC’s effect on this model [16] and the fact that 3K3A-aPC is more effective compared to normal aPC.

### 4.3. Clinical Evaluation of CHS

The clinical severity of CHS was assessed via measurement of the ear thickness at day 0 and day 3, then daily from day 5 to 14. At day 14, mice were euthanized; the ears, blood, inguinal LN, and spleens were harvested for further investigation.

The use of animals was approved by the Northern Sydney Local Health District (NSLHD) Animal Ethics Committee and conducted according to the NSLHD Guide for the Care and Use of Laboratory Animals.

### 4.4. Histological Examination

The flank skin and ear tissues of normal and CHS mice (8–10 weeks old) were fixed in 10% formalin. Tissue sections were stained with H&E and toluidine blue. Mast cells in the skin were identified by metachromasia in toluidine staining and evaluated in 5 random fields per slide under microscopy (400×). All measurements were performed under blinded conditions by two independent researchers.

### 4.5. Immune Cell Detection

Skin tissues were dissociated by Liberase (Merck KGaA, Darmstadt, Germany) digestion. Cell suspensions of skin, blood, LN, and spleen were stained with mouse antibody panels (BD, BD North Ryde, NSW, Australia) consisting of FV510-BV510, CD80-BV605, NK1.1-BV650, CD4-BV711, CD117-BV786, CD11b-BB515, CD19-PerCP/Cy5.5, CD115-PE, Ly6G-PE/CF594, CD3-PE/Cy7, CD206-AF647, CD8a-AF700, and Ly6C-APC/Cy7 to identify immune cells; CD3-AF700, CD4-BV711, T-bet-BV786, GATA3-PECY7, Ror-γ-BV421, FOX3-PE, and CD25-APC to identify Th1/Th2/Th17/Treg cells; CD11C-PE-CF594, CD317-BV421 (BST2), CD45R/B220-APC-H7, CD11b-BB515, IAb-BV711, CD80-BV605, CD86-PE-Cy7, and CD40-APC to identify DCs (CD11c+), myeloid (m)DCs (CD11c+CD11b+B220+), pDCs (CD11c+CD11b-B220+PDCA-1+), cDCs (CD11b+CD11c+), and DC maturation (levels of CD80, CD86, and CD40, I-Ab MHC class II alloantigen (IAb)). Detection was performed using a LSR Fortessa flow cytometer (BD, BD North Ryde, NSW, Australia), and data were analyzed in FlowJo (software version 10.9) (BD, BD North Ryde, NSW, Australia).

### 4.6. Cytokine and IgE Detection

Ear tissues were homogenized and collected into a protein extraction reagent (Thermo Fisher Scientific, Waltham, MA, USA). After vortexing, the homogenates were centrifuged and the clear supernatants collected. IL-4, IL-6, IL-17, IFN-γ, TGF-β1, and TNF-α (R&D Systems, Minneapolis, MN, USA), IL-22 and IgE (Biolegend, San Diego, CA, USA) in tissue supernatants or plasma were measured by enzyme-linked immunosorbent assays (ELISA) according to the manufacturers’ instructions.

### 4.7. aPC Activity

The activity of aPC in the plasma and the skin homogenates was measured by the chromogenic substrate Spectrozyme PCa assay (American Diagnostica Inc., Stamford, CT, USA). aPC activity was determined by measuring the increase in absorbance of the free chromophore generated at 450 nm after acid quenching of the reaction at excess substrate concentrations. The absorbance increases due to the amount of chromophore released are linearly related to aPC concentration. Recombinant human aPC was used to generate a standard curve to calculate aPC activity.

### 4.8. Immunohistochemical Staining

Mouse skin tissue sections were de-paraffinized, heat-retrieved, and immunostained using rabbit anti-mouse PAR1 and goat anti-mouse PAR2 antibodies (Santa Cruz Biotech, Dallas, TX, USA), goat anti-mouse EPCR and rabbit anti-mouse PC/aPC antibodies (R&D Systems). Tissue sections were then incubated with appropriate ready-to-use secondary antibodies, followed by DAB staining, H&E counterstaining, and being photographed. The staining intensity was determined in five random areas at 200× magnification. The intensity scores were 0 (no staining), 1 (weak staining), 2 (moderate), and 3 (intense staining). Anti-rabbit or goat IgG were used as negative controls, which did not show any staining when used at the same concentrations as primary antibodies.

### 4.9. Statistical Analysis

Data were normally distributed, and data comparisons were performed using the two-tailed Student’s *t*-test, Wilcoxon non-parametric test, or one-way ANOVA followed by Student–Newman–Keuls test, as appropriate, in GraphPad Prism 10.0.0 (GraphPad Software, Boston, MA, USA) or Excel. (Microsoft 365, Microsoft Australia, North Sydney, NSW, Australia). Data are expressed as means ± SD. Two-sided *p* < 0.05 was considered statistically significant.

## Figures and Tables

**Figure 1 ijms-25-01255-f001:**
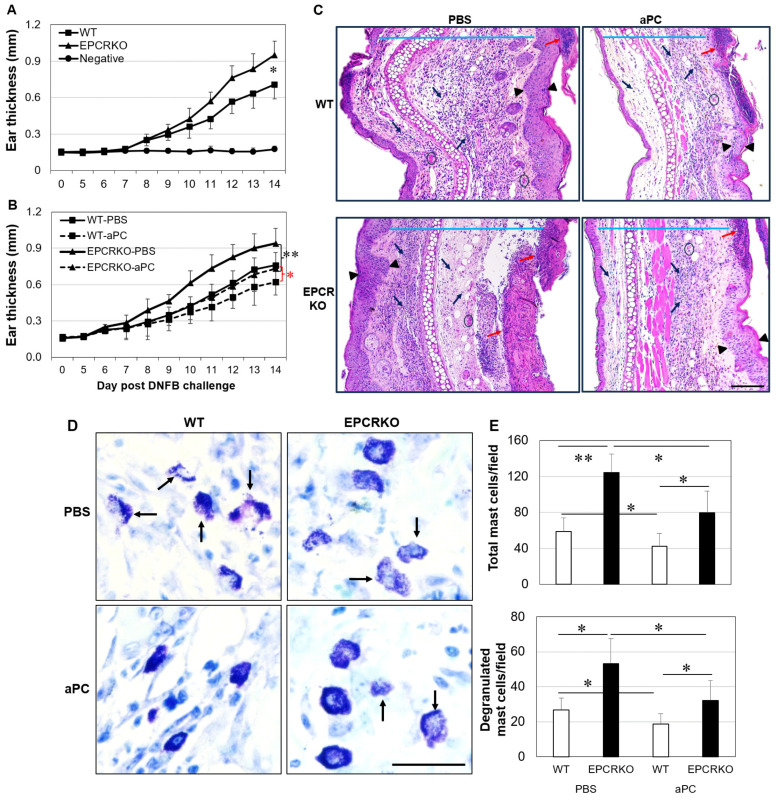
EPCRKO mice exhibited more severe CHS, and 3K3A-aPC inhibited DNFB-induced CHS. (**A**) The ear thickness of WT and EPCRKO mice with CHS or solvent (Negative). (**B**) The ear thickness of mice with CHS treated with phosphate-buffered saline (PBS) (WT-PBS, EPCRKO-PBS) or 3K3A-aPC (WT-aPC, EPCRKO-aPC). (**C**) Representative images of CHS ear skin tissues in WT or EPCRKO mice treated with PBS or 3K3A-aPC (aPC) at day 14, stained with H&E. Scale bar: 200 µm. Blue lines indicate edema of dermis; black arrows indicate the inflammatory infiltrates; red arrows indicate ulceration/erosion. Arrow heads indicate epidermal hyperplasia; black circles indicate hemorrhage. (**D**) The representative images of mast cells in ear skin tissues of CHS, stained with toluidine blue. Black arrows indicate degranulated mast cells. Scale bar: 50 µm. (**E**) Mast cells quantified in 5 random fields per slide (400×) under microscopy. Data shown in the graphs are means ± SD (n = 8 for all groups). * *p* < 0.05, ** *p* < 0.01.

**Figure 2 ijms-25-01255-f002:**
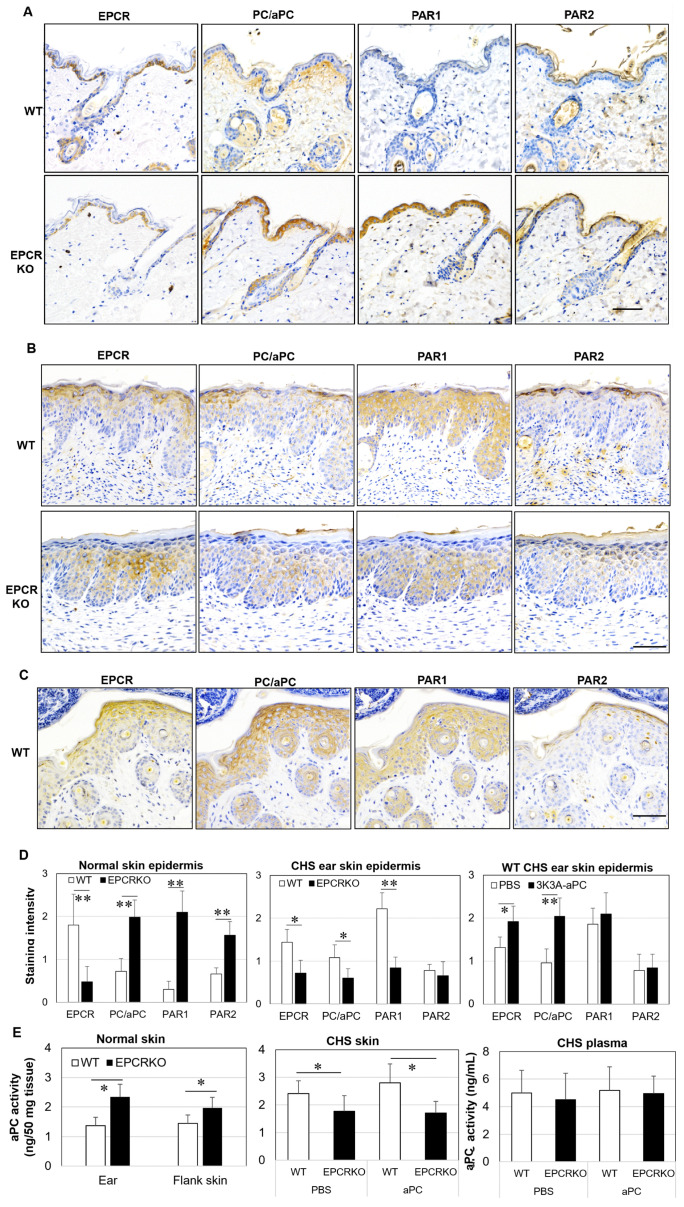
The expression/activity of EPCR, PC/aPC, PAR1, and PAR2 in the skin of normal and CHS mice. (**A**) Representative images of EPCR, PC/aPC, PAR1, and PAR2 expression in skin tissues of WT and EPCRKO mice (8 weeks old, flank skin), detected by immunochemistry. (**B**) Representative images of EPCR, PC/aPC, PAR1, and PAR2 expression in ear tissues of WT and EPCRKO mice with CHS at day 14. (**C**) Representative images of EPCR, PC/aPC, PAR1, and PAR2 expression in skin tissues of WT mice treated with 3K3A-aPC (1 mg/kg). Scale bars in (**A**–**C**): 100 µm. (**D**) The staining intensity of EPCR, PC/aPC, PAR1, and PAR2 in the skin epidermis in (**A**–**C**), expressed as means ± SD (n = 8). (**E**) aPC activity in the ear skin tissues/plasma, expressed as means ± SD (n = 8 for all groups), against human recombinant aPC. * *p* < 0.05. ** *p* < 0.01.

**Figure 3 ijms-25-01255-f003:**
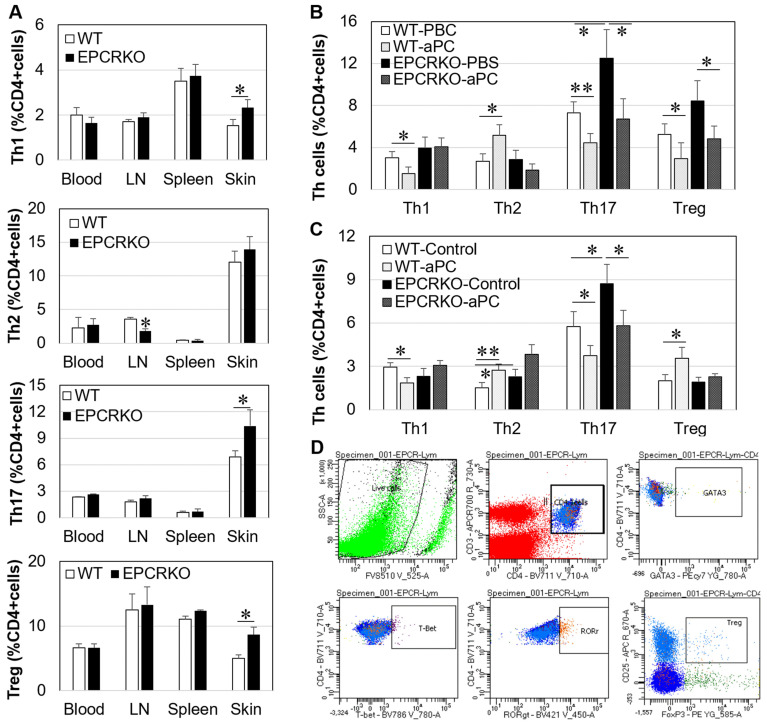
The effect of EPCR and 3K3A-aPC on Th and Treg cells in vivo and in vitro. Cell suspensions of blood, lymph nodes (LN), spleens, or ear skin tissues were obtained from mice with CHS at day 14. Th cell phenotypes were detected by flow cytometry. (**A**) Th1/2/17 and Treg cells in tissues of WT or EPCRKO CHS mice. (**B**) Th1/2/17 and Treg cells in ear skin tissues of CHS mice treated with PBS (WT-PBS, EPCRKO-PBS) or 3K3A-aPC (1 mg/kg, WT-aPC, EPCRKO-aPC). (**C**) Th1/2/17 and Treg cells in WT or EPCRKO spleen cells treated with 3K3A-aPC (10 µg/mL, WT-aPC, EPCRKO-aPC) for 24 h in vitro. (**D**) Gating strategies for Th/Treg cells. Data in the graphs are shown as means ± SD (n = 8 mice or 3 independent in vitro experiments). * *p* < 0.05, ** *p* < 0.01.

**Figure 4 ijms-25-01255-f004:**
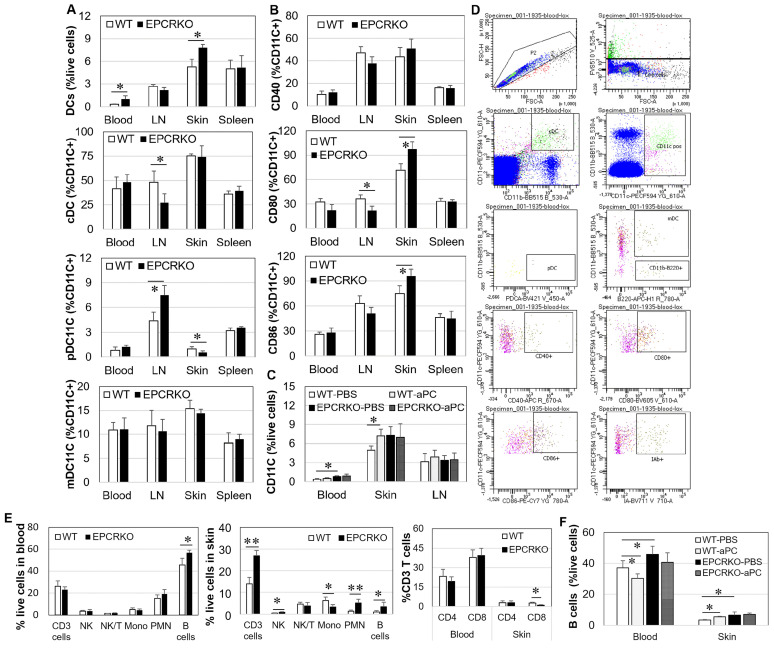
DCs and other immune cells in WT and EPCRKO mice with CHS. (**A**–**C**) DCs and their sub-sets cDCs, plasmacytoid (p)DCs and myeloid (m)DCs (**A**), and their mature status (B) in tissues of WT and EPCRKO mice with CHS or CHS treated with PBS (WT-PBS, EPCRKO-PBS) or 3K3A-aPC (1 mg/kg, WT-aPC, EPCRKO-aPC) (**C**) at day 14. (**D**) Flow cytometric gating strategies for (**A**–**C**). (**E**) CD3 T cells, natural killer cells (NK), NK T cells (NK/T), monocytic cells (Mono), polymorphonuclear leukocytes (PMN), and B cells in blood and ear skin tissues. (**F**) B cells in ear skin tissues of CHS mice treated with PBS or 3K3A-aPC; (**E**,**F**) were detected by flow cytometry. Data in the graphs are shown as means ± SD (n = 4 mice or 3 independent experiments). * *p* < 0.05, ** *p* < 0.01.

**Figure 5 ijms-25-01255-f005:**
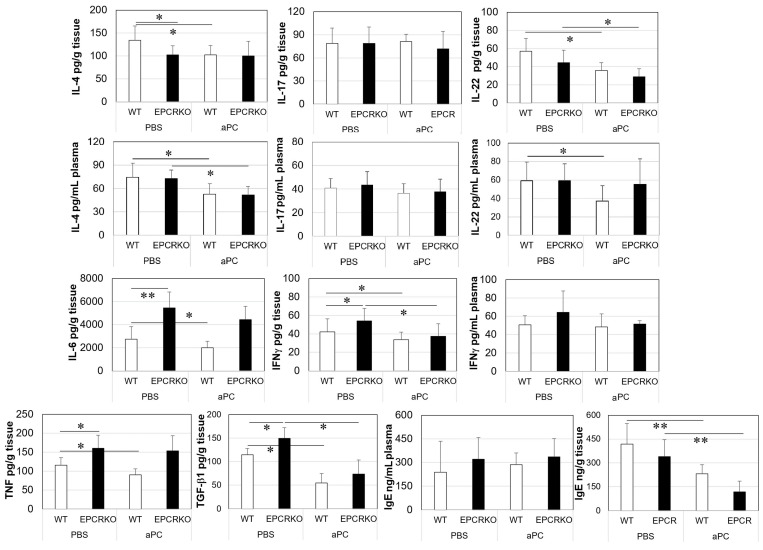
Cytokines and IgE in ear tissues or plasma of mice with CHS. Mice treated with phosphate-buffered saline (PBS) or 3K3A-aPC (aPC) were euthanized at day 14 following the DNFB challenge. Ears and blood were harvested. Plasma was extracted from blood, and ear tissues were homogenized in a protein extract reagent. The tissue homogenates were centrifuged and the clear supernatants collected. Cytokines and IgE in ear tissue supernatants and plasma were measured by enzyme-linked immunosorbent assay. Data shown in the graphs are means ± SD (n = 8 for all groups). * *p* < 0.05, ** *p* < 0.01.

## Data Availability

The data underlying this article will be shared on reasonable request to the corresponding author.

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
