# Peer review of "Endothelial Protein C Receptor and 3K3A-Activated Protein C Protect Mice from Allergic Contact Dermatitis in a Contact Hypersensitivity Model"

_ijms, 2024, doi:10.3390/ijms25021255_

Round 1

Reviewer 1 Report

Comments and Suggestions for Authors

The manuscript "Endothelial protein C receptor protects mice from allergic contact dermatitis in a contact hypersensitivity model" is interesting. However, the study presented is not appropriate and need to address

1. The study is too meshy. Please restructure the document to simplify the study. Numerous unnecessary abbreviations impede the smooth flow and readability of the article. Avoid using abbreviations for basic terms, such as mast cells (MC).

2. The abstract contains an excessive number of abbreviations, rendering it challenging to comprehend. When these abbreviations are initially introduced in the manuscript, provide their full names (e.g., DNFB). Ensure a comprehensive review of the entire manuscript for this correction.

3. If 3K3A-APC (line 23) and 3K3A-aPC (lines 29 and 31) are the same, address the issue in the subsequent section also.

4. Please clarify the intended meaning of keyword 1. Additionally, all the keywords are merely replications of the title. Please consider a more thoughtful selection of keywords.

5. The methodology is relatively moderate. Provide sufficient details so that one can follow these methods. Precisely described in detail in section 4.7.

6. The author performed the parametric One-way ANOVA, specifying that before performing it; did the the author check the normality of the data. Please state it in section 4.9.

7. Rewrite 38-39 in the introduction section.

8. "It can ;…… stemness [7-9] (lines 42-46). Specify it can?

9. "Disruption of … Disruption [13]. Lines 47-51 are contradictory; rewrite it adequately.

10. Define the objective in a more detailed manner.

11. In figure 1C describes the distinct changes using arrows in the histological images.

12. For Figure 1A and B, rewrite the axis as 1…..14. Rewrite the Figure 5 caption to make it more informative.

Comments on the Quality of English Language

Moderate english change required

Author Response

We thank you for your instructive comments. Please see below our response to your comments.

The manuscript "Endothelial protein C receptor protects mice from allergic contact dermatitis in a contact hypersensitivity model" is interesting. However, the study presented is not appropriate and need to address

  1. The study is too meshy. Please restructure the document to simplify the study. Numerous unnecessary abbreviations impede the smooth flow and readability of the article. Avoid using abbreviations for basic terms, such as mast cells (MC).

Response: These issues have been addressed. Please see the tracked changes. Several abbreviations, including mast cells, have been deleted.

  1. The abstract contains an excessive number of abbreviations, rendering it challenging to comprehend. When these abbreviations are initially introduced in the manuscript, provide their full names (e.g., DNFB). Ensure a comprehensive review of the entire manuscript for this correction.

Response: These issues have been addressed. Please see the tracked changes. The full spelling for DNFB has been provided.

  1. If 3K3A-APC (line 23) and 3K3A-aPC (lines 29 and 31) are the same, address the issue in the subsequent section also.

Response: Thank you. We revised all 3K3A-APC to 3K3A-aPC.

  1. Please clarify the intended meaning of keyword 1. Additionally, all the keywords are merely replications of the title. Please consider a more thoughtful selection of keywords.

Response: The keyword 1 endothelial protein C receptor is a specific receptor for the natural anticoagulant, protein C (PC) and its activated form, aPC. We modified the keywords to skin inflammation, mutant activated protein C; immune cells; protease-activated receptor.

The methodology is relatively moderate. Provide sufficient details so that one can follow these methods. Precisely described in detail in section 4.7.

Response: These details have been added to section 4.7: “The activity of aPC in the plasma and the skin homogenates was measured by the chromogenic substrate Spectrozyme PCa assay (American Diagnostica Inc., Stamford, CT, USA). aPC activity was determined by measuring the increase in absorbance of the free chromophore generated at 450 nm after acid quenching of the reaction at excess substrate concentrations. The absorbance increases due to the amount of chromophore released is linearly related to aPC concentration. Recombinant human aPC was used to generate a standard curve to calculate the aPC activity.”

  1. The author performed the parametric One-way ANOVA, specifying that before performing it; did the the author check the normality of the data. Please state it in section 4.9.

Response: Yes, the data were normally distributed, and no outliers were detected. This statement has been added to this section.

  1. Rewrite 38-39 in the introduction section.

Response: Thank you. We revised this sentence as “This condition is characterized by impaired skin barrier and altered immune response 1-3”  (move to line 62-63 in tracked change version).

  1. "It can ;…… stemness [7-9] (lines 42-46). Specify it can?

Response: We added “for being the potential stem cell markers for several types of cells 4 including human keratinocytes5” to this sentence.

  1. "Disruption of … Disruption [13]. Lines 47-51 are contradictory; rewrite it adequately.

Response: These sentences have been modified to: The disruption of EPCR gene expression by embryonic stem cells leads to early embryonic lethality in mice6. Mice with EPCR severe deficiency are generated by the integration of exogenous DNA elements into the 5'-untranslated region of the EPCR gene7. These mice develop normally but are more prone to dextran sulfate sodium-induced colitis, which causes inflammation and mucosal barrier damage8. Conversely, over-expression of EPCR protects transgenic mice from endotoxin-induced injury 9.

  1. Define the objective in a more detailed manner.

Response: This is defined at the end of introduction: “By utilizing EPCR deficient (KO) and matched wild-type (WT) mice in this mouse CHS model, the current study aimed to investigate potential protective effects of EPCR and 3K3A-aPC in this disease.”

  1. In figure 1C describes the distinct changes using arrows in the histological images.

Response: Relevant arrows/illustrations were added to figure 1C.

  1. For Figure 1A and B, rewrite the axis as 1…..14. Rewrite the Figure 5 caption to make it more informative.

Response:  Figure 1A and B, x- axis was rewritten as suggested and more detail was added to figure 5 caption.

References

  1. Martin SF. Induction of contact hypersensitivity in the mouse model. Methods Mol Biol 2013; 961: 325-35.
  2. Nassau S, Fonacier L. Allergic Contact Dermatitis. Med Clin North Am 2020; 104(1): 61-76.
  3. Saint-Mezard P, Berard F, Dubois B, Kaiserlian D, Nicolas JF. The role of CD4+ and CD8+ T cells in contact hypersensitivity and allergic contact dermatitis. Eur J Dermatol 2004; 14(3): 131-8.
  4. Mohan Rao LV, Esmon CT, Pendurthi UR. Endothelial cell protein C receptor: a multiliganded and multifunctional receptor. Blood 2014; 124(10): 1553-62.
  5. Xue M, Dervish S, Chan B, Jackson CJ. The endothelial protein C receptor is a potential stem cell marker for epidermal keratinocytes. Stem cells 2017; 35(7): 1786-98.
  6. Gu JM, Crawley JTB, Ferrell G, et al. Disruption of the Endothelial Cell Protein C Receptor Gene in Mice Causes Placental Thrombosis and Early Embryonic Lethality. Journal of Biological Chemistry 2002; 277(45): 43335-43.
  7. Castellino FJ, Liang Z, Volkir SP, et al. Mice with a severe deficiency of the endothelial protein C receptor gene develop, survive, and reproduce normally, and do not present with enhanced arterial thrombosis after challenge. Thromb Haemost 2002; 88(3): 462-72.
  8. Kondreddy V, Keshava S, Esmon CT, Pendurthi UR, Rao LVM. A critical role of endothelial cell protein C receptor in the intestinal homeostasis in experimental colitis. Sci Rep 2020; 10(1): 20569.
  9. Li W, Zheng X, Gu J, et al. Overexpressing endothelial cell protein C receptor alters the hemostatic balance and protects mice from endotoxin. J ThrombHaemost 2005; 3(7): 1351-9.

Reviewer 2 Report

Comments and Suggestions for Authors

The manuscript entitled “Endothelial protein C receptor protects mice from allergic contact dermatitis in a contact hypersensitivity model” addresses the beneficial effects of the mutant form of activated protein C (3K3A-APC) for the mitigation of contact dermatitis in a mouse model of contact hypersensitivity (CHS). Herein, the authors used wild-type female mice and Endothelial protein C receptor (EPCR)-knockout mice. Endothelial protein C receptor (EPCR). The authors proved that EPCRKO mice demonstrated more severe CHS with elevated immune cell infiltration in the skin and higher levels of inflammatory cytokines and IgE than WT. Interestingly, K3A-aPC reduced the CHS severity in WT and EPCRKO mice by dampening the infiltration/activation of immune cells and pro-inflammatory cytokines. The current findings are interesting, and all sections of the manuscript are written clearly.

Comments:     

1) To avoid readers’ confusion, the current title needs to be modified to fit well with the obtained data. Since the agonist of the EPC receptor is what triggered the beneficial actions against the Endothelial protein C receptor, it should be mentioned directly in the title rather than the name of the receptor (EPCR). Specifically, the mutant form of activated protein C (3K3A-APC) rather than the WT APC (which triggers excessive bleeding tendency) is the agonist used herein, and its name should show up in the title. Please, modify the title accordingly.

2) To enhance the readership of the current research, in the introduction section, the authors are advised to describe the experimental model used in the current study in detail (DNFB-induced contact hypersensitivity model). What are its advantages, and how it resembles human contact dermatitis?

3) In the main text, please explain each abbreviation only once, when used for the first time, then use an abbreviation consequently.

For example, in line 59 (introduction), the abbreviation DNFB was mentioned for the first time without providing its full name. Please, address this point in the entire manuscript. 

4) In the statistical analysis section, did the authors check data normality before proceeding to student-t-test or one-way ANOVA? Authors are advised to address this point and add the answers in the material and methods section.

5) Since the authors mentioned one-way ANOVA in section 4.9., the authors are advised to add the name of the post-hoc test to this section.

6) In section 4.1., it is not clear why the authors chose female mice which are variably affected by the female estrous cycle. This could have potential effects on the pathogenesis of CHS. Please, justify and add proper citations in the material and methods section.

7) In section 4.1., please, add the number of animals used in each experimental group.

8) In section 4.2., how was the dose of 3K3A-aPC (1 mg/kg) selected? Is the selected dose in mice relevant for human translation? Can you discuss the dose used for possible translation in humans, for example, by using conversion tables available in the literature using the Human effective dose (HED) formula= animal dose x animal Km/ human Km (Nair AB, Jacob S. A simple practice guide for dose conversion between animals and humans. J Basic Clin Pharm. 2016 Mar;7(2):27-31). I would suggest that authors address this point and add the answers/proper citations to section 4.2.

9) In histopathology (Figure 1C), please use higher magnification. The authors are advised to describe whether edema, hemorrhage, etc. occurred in the tissues. The different pathological changes in the histology images should be pointed at by arrows (with different shapes, if needed), properly labeled, and described in the figure legend properly.

10) In immunohistochemistry (section 4.8.), did the authors also perform a negative control to ensure the specific binding of antibody to target protein? Please, add the answer to the comment in section 4.8.

11) In Figure 2A-C, please quantify the expression of these proteins. 

Author Response

We thank you for your instructive comments. Please see below our response to your comments.

The manuscript entitled “Endothelial protein C receptor protects mice from allergic contact dermatitis in a contact hypersensitivity model” addresses the beneficial effects of the mutant form of activated protein C (3K3A-APC) for the mitigation of contact dermatitis in a mouse model of contact hypersensitivity (CHS). Herein, the authors used wild-type female mice and Endothelial protein C receptor (EPCR)-knockout mice. Endothelial protein C receptor (EPCR). The authors proved that EPCRKO mice demonstrated more severe CHS with elevated immune cell infiltration in the skin and higher levels of inflammatory cytokines and IgE than WT. Interestingly, K3A-aPC reduced the CHS severity in WT and EPCRKO mice by dampening the infiltration/activation of immune cells and pro-inflammatory cytokines. The current findings are interesting, and all sections of the manuscript are written clearly.

Comments:     

  • To avoid readers’ confusion, the current title needs to be modified to fit well with the obtained data. Since the agonist of the EPC receptor is what triggered the beneficial actions against the Endothelial protein C receptor, it should be mentioned directly in the title rather than the name of the receptor (EPCR). Specifically, the mutant form of activated protein C (3K3A-APC) rather than the WT APC (which triggers excessive bleeding tendency) is the agonist used herein, and its name should show up in the title. Please, modify the title accordingly.

Response: The title has been modified as suggested to “Endothelial protein C receptor and 3K3A-activated protein C protect mice from allergic contact dermatitis in a contact hypersensitivity model”.

2) To enhance the readership of the current research, in the introduction section, the authors are advised to describe the experimental model used in the current study in detail (DNFB-induced contact hypersensitivity model). What are its advantages, and how it resembles human contact dermatitis?

Response: The last paragraph of introduction has been modified to “The subacute and subchronic 2,4-dinitrofluorobenzene (DNFB)-induced mouse CHS model has been shown to better reflect the morphological and functional specifics of chronic ACD. Moreover, this model allows the therapeutic treatment of a pre-established stable and non-necrotic skin inflammation 10.”.

  • In the main text, please explain each abbreviation only once, when used for the first time, then use an abbreviation consequently.

For example, in line 59 (introduction), the abbreviation DNFB was mentioned for the first time without providing its full name. Please, address this point in the entire manuscript. 

Response: This issue has been extensively examined and revised accordingly. The full spelling for DNFB has been provided, e.g. 2,4-dinitrofluorobenzene.

  • In the statistical analysis section, did the authors check data normality before proceeding to student-t-test or one-way ANOVA?Authors are advised to address this point and add the answers in the material and methods section.

Response: Yes, we checked the data normality and there were no outliers. This statement was added to the material and methods section.

  • Since the authors mentioned one-way ANOVA in section 4.9., the authors are advised to add the name of the post-hoc test to this section. 

Response: One-way ANOVA followed by a Student–Newman–Keuls test has been added to this section.

  • In section 4.1., it is not clear why the authors chose female mice which are variably affected by the female estrous cycle. This could have potential effects on the pathogenesis of CHS. Please, justify and add proper citations in the material and methods section.

Response: Female estrous cycle has no significant impact on this model and CHS can be induced in both male and female mice11. The female mice are commonly used in this model owing to their less aggressive behavior. This information was added to this section.

  • In section 4.1., please, add the number of animals used in each experimental group. 

Response: The number of mice/group was added to section 4.1.

  • In section 4.2., how was the dose of 3K3A-aPC (1 mg/kg) selected? Is the selected dose in mice relevant for human translation? Can you discuss the dose used for possible translation in humans, for example, by using conversion tables available in the literature using the Human effective dose (HED) formula= animal dose x animal Km/ human Km (Nair AB, Jacob S. A simple practice guide for dose conversion between animals and humans. J Basic Clin Pharm. 2016 Mar;7(2):27-31). I would suggest that authors address this point and add the answers/proper citations to section 4.2.

Response: Thank you for sharing this very useful reference with us. This dose of 3K3A-aPC (1 mg/kg) chosen was based on our previous study which showed that human recombinant APC at 2 mg/kg was effective in the treatment of CHS in the same mouse model12. However, in our recent study using mouse and pig wound healing models and in vitro skin cell models, we discovered that a lower dose of 3K3A-aPC (approximately half of APC dose) was better than the equivalent dose of APC13. Therefore, we selected 3K3A-aPC at 1 mg/kg for our study. According to the literature you provided, the relevant dose for possible translation in humans would be 12.3mg/kg. This information will serve a useful guideline for the future clinical trials.

  • In histopathology (Figure 1C), please use higher magnification. The authors are advised to describe whether edema, hemorrhage, etc. occurred in the tissues. The different pathological changes in the histology images should be pointed at by arrows (with different shapes, if needed), properly labeled, and described in the figure legend properly.

Response: Figure 1C has been revised as suggested. Please see the figure caption for detail.

  • In immunohistochemistry (section 4.8.), did the authors also perform a negative control to ensure the specific binding of antibody to target protein? Please, add the answer to the comment in section 4.8.

Response: Yes, anti-rabbit and goat IgG were used as negative controls, which did not show any staining when used at the same concentrations as primary antibodies. This information has been added to this section.

  • In Figure 2A-Cplease quantify the expression of these proteins. 

Response: EPCR, PC/APC, PAR1 and PAR2 immunohistochemical staining results have been semi-quantified, and the relevant graphs were added into Figure 2 as “ D) The staining intensity of EPCR, PC/APC, PAR1 and PAR2 in the skin epidermis from A-C, expressed as means ± SD (n = 8).” The semi-quantification method was added to materials and methods section 4.8.

References:

  1. Rose L, Schneider C, Stock C, Zollner TM, Docke WD. Extended DNFB-induced contact hypersensitivity models display characteristics of chronic inflammatory dermatoses. Exp Dermatol 2012; 21(1): 25-31.
  2. Manresa MC. Animal Models of Contact Dermatitis: 2,4-Dinitrofluorobenzene-Induced Contact Hypersensitivity. Methods Mol Biol 2021; 2223: 87-100.
  3. Xue M, Lin H, Liang HPH, et al. Deficiency of protease-activated receptor (PAR) 1 and PAR2 exacerbates collagen-induced arthritis in mice via differing mechanisms. Rheumatology (Oxford) 2021; 60(6): 2990-3003.
  4. Zhao R, Xue M, Lin H, et al. A recombinant signalling-selective activated protein C that lacks anticoagulant activity is efficacious and safe in cutaneous wound preclinical models. Wound Repair Regen 2023.

Reviewer 3 Report

Comments and Suggestions for Authors

The manuscript “Endothelial protein C receptor protects mice from allergic contact dermatitis in a contact hypersensitivity model” may be useful in the understanding of the mechanisms of allergic contact dermatitis and in the treatment patients with this disease. I would like to make a few comments:

1.       Lines 39-40: “The exact mechanisms of ACD are not fully understood.”

The pathogenesis of ACD is well described, read for example:

Yamaguchi, H.L.; Yamaguchi, Y.; Peeva, E. Role of Innate Immunity in Allergic Contact Dermatitis: An Update. Int. J. Mol. Sci. 2023, 24, 12975. https://doi.org/10.3390/ijms241612975

Murphy PB, Atwater AR, Mueller M. Allergic Contact Dermatitis. [Updated 2023 Jul 13]. In: StatPearls [Internet]. Treasure Island (FL): StatPearls Publishing; 2023 Jan-. Available from: https://www.ncbi.nlm.nih.gov/books/NBK532866/

2.       Indicate, please the aim of the study in the introduction. Replace the text on lines 59-61 to the Results.

3.       Please delete the word “Single” in the Figure 3, the sentence “Single cell suspensions of blood, lymph nodes (LN), spleens, or ear skin tissues were obtained…”

The term “Single cell suspension” is associated with “single cell” techniques that are not relevant to this study.

4.       In the figure 3 the authors provide explanations under the A, B and C letters, but missed under the letter D. It is necessary to add a caption to the figure 3.

5.       In the item Materials and Methods indicate please the manufacturers of all reagents, including 3K3A-aPC.

Author Response

We thank you for your instructive comments. Please see below our response to your comments. 

Comments and Suggestions for Authors

The manuscript “Endothelial protein C receptor protects mice from allergic contact dermatitis in a contact hypersensitivity model” may be useful in the understanding of the mechanisms of allergic contact dermatitis and in the treatment patients with this disease. I would like to make a few comments:

   Lines 39-40: “The exact mechanisms of ACD are not fully understood.”

The pathogenesis of ACD is well described, read for example:

 Yamaguchi, H.L.; Yamaguchi, Y.; Peeva, E. Role of Innate Immunity in Allergic Contact Dermatitis: An Update. Int. J. Mol. Sci. 2023, 24, 12975. https://doi.org/10.3390/ijms241612975

 Murphy PB, Atwater AR, Mueller M. Allergic Contact Dermatitis. [Updated 2023 Jul 13]. In: StatPearls [Internet]. Treasure Island (FL): StatPearls Publishing; 2023 Jan-. Available from: https://www.ncbi.nlm.nih.gov/books/NBK532866/

Response: This sentence has been changed to “Although the mechanisms of ACD are well defined 14, the function of EPCR in ACD is unknown. “

  1. Indicate, please the aim of the study in the introduction. Replace the text on lines 59-61 to the Results.

Response: This was modified as suggested to “By utilizing EPCR deficient (KO) and matched wild-type (WT) mice in this mouse CHS model, the current study aimed to investigate potential protective effects of EPCR and 3K3A-aPC in this disease.”

  1. Please delete the word “Single” in the Figure 3, the sentence “Single cell suspensions of blood, lymph nodes (LN), spleens, or ear skin tissues were obtained…”

The term “Single cell suspension” is associated with “single cell” techniques that are not relevant to this study.

Response: In this study, “Single cell suspension” means individual cells that are not associated with each other. To make it clear, we deleted the single.

  1. In the figure 3 the authors provide explanations under the A, B and C letters, but missed under the letter D. It is necessary to add a caption to the figure 3.

Response:  Thank you. This is added to Figure 3. D) Flow cytometric gating strategies for A-C.

  1. In the item Materials and Methods indicate please the manufacturers of all reagents, including 3K3A-aPC.

Response: All associated manufacturers were added.

References:

  1. Yamaguchi HL, Yamaguchi Y, Peeva E. Role of Innate Immunity in Allergic Contact Dermatitis: An Update. Int J Mol Sci 2023; 24(16).

Round 2

Reviewer 1 Report

Comments and Suggestions for Authors

The manuscript has been substantially revised based on the raised comments. I endorse the article for the publication. 

Comments on the Quality of English Language

NA

Reviewer 2 Report

Comments and Suggestions for Authors

The authors have adequately addressed the raised comments; thanks!